

# Inverse modelling for predicting both water and nitrate movement in a structured-clay soil (Red Ferrosol)

James M. Kirkham[1], Christopher J. Smith[2], Richard B. Doyle[1] and Philip H. Brown[3]

[1] Tasmanian Institute of Agriculture, University of Tasmania, Hobart, TAS, Australia
[2] Land and Water, CSIRO, Canberra, ACT, Australia
[3] Centre for Plant and Water Science, Central Queensland University, Bundaberg, QLD, Australia

Corresponding author
Richard B. Doyle,
Richard.Doyle@utas.edu.au

## ABSTRACT

Soil physical parameter calculation by inverse modelling provides an indirect way of estimating the unsaturated hydraulic properties of soils. However many measurements are needed to provide sufficient data to determine unknown parameters. The objective of this research was to assess the use of unsaturated water flow and solute transport experiments, in horizontal packed soil columns, to estimate the parameters that govern water flow and solute transport. The derived parameters are then used to predict water infiltration and solute migration in a repacked soil wedge. Horizontal columns packed with Red Ferrosol were used in a nitrate diffusion experiment to estimate either three or six parameters of the van Genuchten–Mualem equation while keeping residual and saturated water content, and saturated hydraulic conductivity fixed to independently measured values. These parameters were calculated using the inverse optimisation routines in Hydrus 1D. Nitrate concentrations measured along the horizontal soil columns were used to independently determine the Langmuir adsorption isotherm. The soil hydraulic properties described by the van Genuchten–Mualem equation, and the $NO_3^-$ adsorption isotherm, were then used to predict water and $NO_3^-$ distributions from a point-source in two 3D flow scenarios. The use of horizontal columns of repacked soil and inverse modelling to quantify the soil water retention curve was found to be a simple and effective method for determining soil hydraulic properties of Red Ferrosols. These generated parameters supported subsequent testing of interactive flow and reactive transport processes under dynamic flow conditions.

## INTRODUCTION

Simulation models are useful for examining water and solute movement in soil profiles, such as when improving water and nutrient use efficiency or designing fertigation systems (*Cote et al., 2003*; *Skaggs et al., 2004*; *Siyal & Skaggs, 2009*). There are a number of soil water models, such as LeachM (*Wagenet & Hutson, 1989*), Wet-Up (*Cook et al., 2003*),

Hydrus 1D (*Šimůnek et al., 2008*), Hydrus 2D/3D (*Šimůnek, Van Genuchten & Šejna, 2006*), and numerical procedures described by *Wu & Chieng (1995a, 1995b)* which are all capable of describing water flow, and in some cases solute transport, in one, two, or three dimensions. In this study, we selected the suite of Hydrus models, because they can simulate solute flow under both 1D and 3D conditions. The van Genuchten–Mualem water content, capillary pressure and hydraulic conductivity models were used to predict water flow (*Šimůnek, Van Genuchten & Šejna, 2006*), but physically realistic parameters are needed for the intended application if accurate predictions are to be made.

Inverse optimisation techniques have become increasingly popular for parameter estimation and many soil models now have user-friendly optimisation tools built in (*Hopmans et al., 2002*; *Vrugt & Bouten, 2002*; *Wöhling, Vrugt & Barkle, 2008*; *Kandelous et al., 2011*). The method involves multiple calculations in which parameters are adjusted, using a method such as the Levenberg–Marquardt or Bayesian procedure, until predictions agree sufficiently well with the measured data (*Šimůnek, Van Genuchten & Šejna, 2006*). This has advantages over other techniques for estimating hydraulic parameters, such as pedotransfer functions (PTFs), because the optimised parameters are estimated directly from measured data for a particular soil hydrological problem of interest. Care however must be taken when using this method to ensure parameters are physically realistic and representative of the spatial scale of interest (*Hopmans et al., 2002*; *Vrugt & Bouten, 2002*; *Mallants et al., 2007*; *Wöhling, Vrugt & Barkle, 2008*).

*Šimůnek et al. (2000)* used inverse optimisation to estimate soil hydraulic parameters from water content data measured in horizontal absorption columns. Similarly, inverse optimisation has been used in Hydrus to predict water flow from water potential and cumulative outflow data (*Van Dam, Stricker & Droogers, 1994*; *Hopmans et al., 2002*; *Arbat et al., 2008*). *Kandelous & Šimůnek (2010a, 2010b)* and *Kandelous et al. (2011)* used inverse optimisation to estimate soil hydraulic parameters to predict water distribution from a point source, including subsurface irrigation, in the field. *Mallants et al. (2007)* used Hydrus 2D and cumulative infiltration data from a deep borehole infiltration test in clayey gravel and carbonated loess soil to estimate field-scale soil hydraulic properties.

Despite the increasing popularity of inverse optimisation, there are few published examples in which parameters derived from unsaturated flow absorption columns have been tested in 3D flow scenarios. Obtaining parameters for a specific flow scenario does not guarantee they will be suitable for extrapolation outside the measured data set to which they were fitted (*Sonnleitner, Abbaspour & Schulin, 2003*). *Vrugt & Bouten (2002)* and *Wöhling, Vrugt & Barkle (2008)* recommend the use of the Metropolis algorithm to determine parameter uncertainty, given measurement errors and the models inability to perfectly represent the system. However, if the derived parameters can be shown to be capable of approximating the observed water content distribution under contrasting conditions, it is likely they are physically realistic for the conditions being investigated. To this end, *Sonnleitner, Abbaspour & Schulin (2003)* and *Kandelous et al. (2011)* used inverse parameter estimation to improve simulations of water content data under different flow scenarios. Minimising the number of optimised parameters, increases the likelihood that the parameters are physically realistic (*Hopmans et al., 2002*).

Although several numerical models, including Hydrus (*Hanson, Šimůnek & Hopmans, 2006*), have looked at reactive solute transport (*Molinero et al., 2008*; *Kuntz & Grathwohl, 2009*; *Nakagawa et al., 2010*), validation of the models has been limited. *Phillips (2006)* used Hydrus and some unpublished data to predict the transport of $K^+$ in unsaturated repacked horizontal columns of reactive soil similar to the one used in this study. In field-scale simulations using Hydrus 1D, *Persicani (1995)* and *Moradi, Abbaspour & Afyuni (2005)* had limited success in simulating reactive metal movement over extended time-scales. However, *Rassam & Cook (2002)* were able to use modelling of solute fluxes in soils to explain results from the field and laboratory measurements of *Rassam, Cook & Gardner (2002)*. Recently, *Ramos et al. (2011, 2012)* provide examples where Hydrus was successfully used to predict water and solute movement under saline conditions. Validation of the reactive solute module in Hydrus has received considerable attention; however a continued effort is needed to demonstrate its ability to properly investigate soil hydrological processes and reactive transport.

In this paper, we use inverse parameter estimation to determine soil hydraulic properties from measured water content profiles in horizontal soil columns (1D transport), and apply the parameters to predicting water flow from a point source into a wedge of soil. We also investigated $NO_3^-$ transport, using an adsorption isotherm determined in the horizontal soil columns that were subsequently used in Hydrus 2D/3D to predict $NO_3^-$ distributions in the soil wedge under two different irrigation scenarios.

## MATERIALS AND METHODS

Experiments used surface soil (0–15 cm) of a free-draining, well-structured Red, Mesotrophic, Humose, Ferrosol (*Isbell, 1996*). Soil was collected from Moina in northwest Tasmania, Australia (41°29′28.80″S and 14°60′34.70″E) from a site under long-term pasture. Samples of soil were air-dried at 40 °C, sieved to retain the <2 mm fraction and stored for later use. Chemical and physical properties are presented in Table 1.

Soil pH and electrical conductivity (EC) were measured on 1:5 soil to water extracts (*Raymond & Higginson, 1992*). Solution concentrations were measured from soil samples wet to a water content of 0.55 $g_w\ g^{-1}$ soil. The solution was extracted by centrifuging samples with 10 $cm^3$ of 1,1,2-trichloro-1,2,2-trifluoroethane (TFE) as described by *Phillips & Bond (1989)*. Exchangeable cations were determined by extraction with 1M $NH_4Cl$ after the water-soluble ions had been extracted. Organic carbon was analysed using the *Walkley & Black (1934)* method. Particle size analysis (United States Department of Agriculture (USDA); *Gee & Bauder, 1986*) was undertaken by pipette method after pretreatment to remove both organic carbon and iron oxides (FeO) using hydrogen peroxide and sodium dithionate, respectively (*McKenzie, Coughlan & Cresswell, 2002*). Semiquantitative mineralogy was determined using X-ray diffraction on the pretreated clay fraction from the particle size analysis.

### Horizontal solute absorption

Absorption of a $NO_3^-$ solution by the soil was measured in horizontal columns between 17 and 50 cm in length depending on absorption periods. The air dry soil was moistened

**Table 1 Soil chemical and physical properties for the surface soil (0–15 cm depth).**

| | | | | | |
|---|---|---|---|---|---|
| pH | 5.8 | | | | |
| EC (mS cm$^{-1}$) | 0.10 | | | | |
| Soil solution cations | Ca | K | Mg | Na | NH$_4$-N |
| ($\mu$mol$_c$ cm$^{-3}$ soil solution) | 46.5 | 19.96 | 11.04 | 18.32 | 13.33 |
| Soil solution anions | NO$_3$–N | Cl | PO$_4$–P | SO$_4$–S | |
| ($\mu$mol$_c$ cm$^{-3}$ soil solution) | 45.84 | 15.07 | 0.52 | 2.90 | |
| Exchangeable cations | Ca | K | Mg | Na | |
| ($\mu$mol$_c$ g$^{-1}$ soil) | 827.46 | 114.17 | 83.34 | 0 | |
| Organic carbon (%) | 4.73 | | | | |
| Particle size distribution (%) | sand | silt | clay | | |
| OC removed | 80 | 12 | 8 | | |
| OC and iron oxides removed | 45 | 22 | 33 | | |
| Clay mineralogy (%) | Quartz | amorphous | Kaolinite, organic | Garnet, Gibbsite | Epidote | Smectite, Rutile, Amphibole |
| | 25–35 | 15–25 | 10–15 | 5–10 | 2–5 | <5 |

before packing into the columns. Columns were packed (using a drop hammer) with relatively dry soil (water content 0.15 g g$^{-1}$) in two to three g increments to achieve a bulk density of close to 1.03 g cm$^{-3}$, which is similar to that measured in the field. Using a Mariotte bottle, a 110 $\mu$mol$_c$ cm$^{-3}$ NO$_3^-$ solution was applied to the inlet of the soil column at zero suction. The outlet of the column remained open, tamped with cotton wool to hold the soil in place. Flow was stopped at set times and the column divided into sections that ranged from 1 to 2.5 cm. Short sections were place near the wetting front to provide an accurate measure of the solute and water contents in this area. The soil sections were transferred to tubes and weighed to determine moist weight.

Two types of column experiments were conducted, and are referred to as Set A and Set B. Each individual experiment in both Set A and Set B used a freshly prepared soil column. Set A consisted of five experiments with infiltration times of 26, 30, 43, 47, and 70 min. In these experiments, water-soluble NO$_3^-$ vs distance in the column was determined by adding deionised water to each column section to make a soil-to-water ratio of 1:5.5 (SD ± 0.4). The soil plus water was weighed. Samples were shaken for 4 h in an end-over-end shaker, centrifuged at 9,800 m s$^{-2}$ for 10 min, and the supernatant decanted and weighed. The soil remaining in the tubes was also weighed.

Set B involved four columns with absorption times for two of 80 and 320 min for the others. Duplicate columns in this series were either (i) extracted as in Set A or (ii) the soil solution was extracted using the TFE method described by *Phillips & Bond (1989)*.

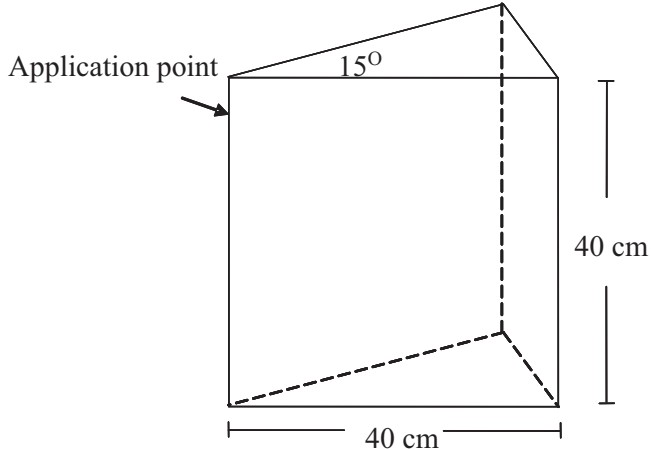

Application point   15°   40 cm   40 cm

**Figure 1  Geometry of the wedge apparatus.**     

The adsorbed $NO_3^-$ was extracted by adding a volume of 2M KCl to form a 1:5.5 (SD ± 0.5) soil:solution ratio (*Rayment & Higginson, 1992*).

The tubes were reweighed and shaken for 1 h to extract adsorbed $NO_3^-$. The soil plus 2M KCl samples were centrifuged at 9,800 m s$^{-2}$ for 10 min, and the supernatant decanted in preweighed falcon tubes and weighed. The soil remaining in the tubes was washed twice by shaking for 30 min in 20 cm$^3$ deionised water to remove residual salts. The tubes were centrifuged at 9,800 m s$^{-2}$ for 10 min and the wash solution discarded. The washed soil was oven-dried at 105 °C and weighed to give the oven-dry mass of soil in each section. Water and KCl extracts were analysed for $NO_3^-$–N on an Alpkem autoanalyser (*Alpkem, 1992*).

## Point-source solute infiltration

Perspex wedges were constructed with the same dimensions described by *Li, Zhang & Ren (2003)* and packed with dry soil (water content of 0.2 g g$^{-1}$) to a bulk density of 0.95 g cm$^{-3}$. All solutions were applied to the 15° corner of the wedge at a depth of five cm (Fig. 1) with a peristaltic pump set to deliver solution at 50 cm$^3$ h$^{-1}$, equivalent to a dripper output of 1,200 cm$^3$ h$^{-1}$ in a 360° flow environment.

Figure 2 shows the two irrigation scenarios applied to the wedge experiments. Both treatments were irrigated for 0.5 h (equivalent to 25 cm$^3$ of solution) with the $NO_3^-$ solution applied to the horizontal columns, that is 110 μmol$_c$ $NO_3^-$ cm$^{-3}$. This was immediately followed by a 1.5 h application of solute free water (75 cm$^3$). The soil was either (i) sampled immediately after the water application (Scenario A) or (ii) allowed to rest for 16 h before irrigating again with water for 6 h (300 cm$^3$) prior to sampling (Scenario B). That is, a total of 375 cm$^3$ of water was applied to the wedge in Scenario B with samples being taken 24 h after initial application of the solute. A single replicate was used for Scenario A and Scenario B was duplicated.

Soil was sampled using the method described by *Li, Zhang & Ren (2003)*. Briefly, a five cm grid was placed over the column and a soil core (two cm internal diameter) taken from the centre of each grid. Additional soil samples were taken at the edge of the wetting front. The soil from the core was sub-sampled to determine gravimetric water content
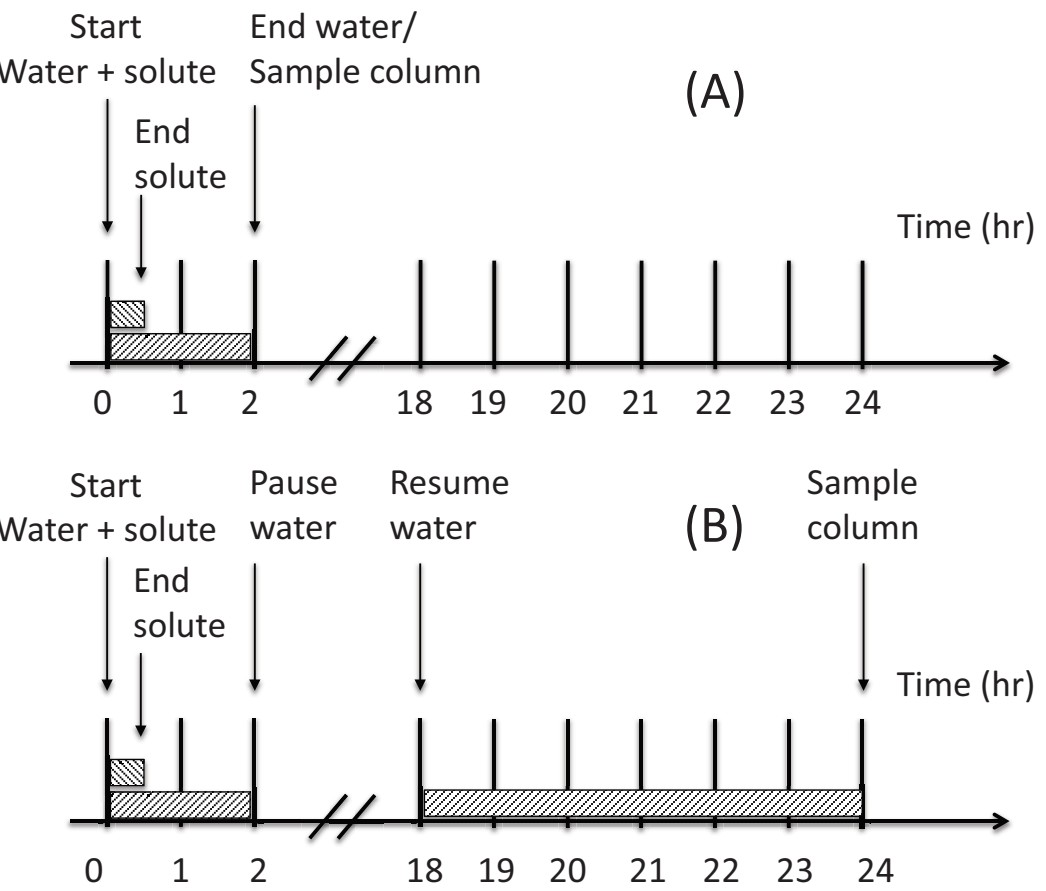

**Figure 2 Schematic of the two irrigation scenarios (A) and (B) applied to the wedge columns.**

and total $NO_3^-$ concentration (solution and adsorbed). Wet soil was extracted with 2M KCl (1:10 soil: KCl ratio) and analysed on an Alpkem autoanalyser (*Rayment & Higginson, 1992*). Calculations of total $NO_3^-$ partitioned into the solution and the amount of adsorbed $NO_3^-$ were done using the adsorption isotherm determined from the horizontal column data (described below).

## Nitrate adsorption isotherm

Partitioning of $NO_3^-$ between the adsorbed and solution phases, over the range of soil solution concentrations in the columns, was measured by displacing the soil solution with TFE (*Phillips & Bond, 1989*). The Langmuir equation (Eq. (1)) was then fitted to the data to describe the $NO_3^-$ adsorption isotherm.

$$C_a = \frac{C_{max} \, \phi \, C_w}{1 + \phi \, C_w},$$ (1)

where $C_a$ is the concentration of adsorbed solute ($\mu mol_c \, g^{-1}$), $C_w$ is the concentration of solute in the soil solution ($\mu mol_c \, cm^{-3}$), $C_{max}$ is the maximum amount of solute that can be adsorbed by the soil ($g \, cm^{-3}$), and $\phi$ determines the magnitude of the initial slope of the isotherm (*Sposito, 1989*).

$C_{\mathrm{max}}$ and $\phi$ were determined using the Gauss–Newton nonlinear curve model in the statistical program SAS (version 9.1; SAS, Cary, NC, USA). $C_{\mathrm{max}}$ and $\phi$ were determined to be 23.17 (95% confidence interval (CI) ± 3.43) and 0.00766 (95% CI ± 0.00194), respectively.

## Hydrus flow equations

Water flow is described by the Richards equation modified to describe horizontal flow in one dimension with no loss of water due to evaporation of root uptake (*Šimůnek et al., 2008*):

$$\frac{\partial \theta}{\partial t} = \frac{\partial}{\partial x}\left( K \frac{\partial \psi}{\partial x} \right),$$

where $\theta$ is the volumetric water content ($\mathrm{cm}^3\ \mathrm{cm}^{-3}$), $t$ is time (min), $x$ is the horizontal distance (cm), $\psi$ is the water tension (cm), and $K$ is the hydraulic conductivity ($\mathrm{cm\ min}^{-1}$) given by:

$$K(\psi, x) = K_{\mathrm{sat}}(x)K_r(h, x),$$

where $K_r$ is the relative hydraulic conductivity (no unit) and $K_{\mathrm{sat}}$ is the saturated hydraulic conductivity ($\mathrm{cm\ min}^{-1}$; *Šimůnek et al., 2008*).

The modified form of the Richards equation that describes water movement in two dimensions assuming no loss of water through root uptake or evaporation can be written (*Šimůnek, Van Genuchten & Šejna, 2006*):

$$\frac{\partial \theta}{\partial t} = \frac{\partial}{\partial x_i}\left[ K\left( K_{ij}^A \frac{\partial \psi}{\partial x_j} \right) + K_{iz}^A \right]$$

where $x_i$ ($i = 1,2$) are the spatial coordinates (cm), and $K_{ij}^A$ and $K_{iz}^A$ are components of a dimensionless anisotropy tensor $K^A$. Assuming flow is isotropic (i.e. $K$ is equal in horizontal and vertical directions) the diagonal entries of $K_{ij}^A$ equal one and the off-diagonal entries equal zero (*Šimůnek, Van Genuchten & Šejna, 2006*).

In the two-dimensional flow scenario $K$ is given by:

$$K(\psi, x) = K_{\mathrm{sat}}(x, z)K_r(h, x, z).$$

If the modified form of the Richards equation is applied to planar flow in a vertical cross section, $x_1 = x$ is the horizontal coordinate and $x_2 = z$ is the vertical coordinate. This equation can also describe axisymmetric flow when $x_1 = x$ represents a radial coordinate (*Gardenas et al., 2005*). The transcripts $i$ and $j$ denote either the $x$ or $z$ coordinate.

To solve the Richards equation, Hydrus implements the soil hydraulic functions of the van Genuchten–Mualem to describe unsaturated hydraulic conductivity in terms of soil water retention parameters (*Šimůnek, Van Genuchten & Šejna, 2006*). Water retention is described by *Šimůnek, Van Genuchten & Šejna (2006)* as:

$$\theta(\psi) = \begin{cases} \theta_r \dfrac{\theta_s - \theta_r}{[1 + |\alpha\psi|^n]^m} & \psi < 0 \\ \theta_s & \psi \geq 0 \end{cases}$$

and unsaturated conductivity is written (*Šimůnek, Van Genuchten & Šejna, 2006*):

$$K(\psi) = K_s S_e^l \left[ 1 - \left( 1 - S_e^{1/m} \right)^m \right]^2 \qquad \psi < 0$$

where:

$$m = 1 - 1/n, \qquad n > 1$$

and:

$$S_e = \frac{\theta - \theta_r}{\theta_s - \theta_r}$$

In the above equations $\theta_r$ is the residual water content ($cm^3\ cm^{-3}$), $\theta_s$ is the saturated water content ($cm^3\ cm^{-3}$), $\alpha$ ($cm^{-1}$), $n$ (no unit), and $l$ (no unit) are curve fitting parameters for the hydraulic conductivity function and $S_e$ is the effective water content ($cm^3\ cm^{-3}$).

When the van Genuchten–Mualem model is used to solve Richards equation in Hydrus there are six soil hydraulic parameters required ($\theta_r$, $\theta_s$, $\alpha$, $n$, $l$, and $K_{sat}$).

The Langmuir equation was used to predict the reactive solute transport. In Hydrus, desorption of a nontransforming solute is described by the generalised nonlinear equation (*Šimůnek, Van Genuchten & Šejna, 2006*):

$$C_a = \frac{k_s C_w^\omega}{1 + \phi C_w^\omega}, \tag{2}$$

and:

$$\frac{\partial C_a}{\partial t} = \frac{k_s \omega C_w^{\omega-1}}{(1 + \phi C_w^\omega)^2} \frac{\partial C_w}{\partial t} + \frac{C_w^\omega}{1 + \phi C_w^\omega} \frac{\partial k_s}{\partial t} - \frac{k_s C_w^{2\omega}}{(1 + \phi C_w^\omega)^2} \frac{\partial \phi}{\partial t} + \frac{k_s C_w^\omega \ln C_w}{(1 + \phi C_w^\omega)^2} \frac{\partial \omega}{\partial t}, \tag{3}$$

where $k_s$ ($cm^3\ g^{-1}$), $\omega$ (dimensionless), and $\phi$ ($cm^3\ g^{-1}$) are constants. In the case of the Langmuir equation, $k_s = C_{max}\ \phi$ (where $C_{max}$ and $\phi$ are the Langmuir equation constants from Eq. (1)) and $\omega = 1$.

## Estimation of flow equations parameters

Soil hydraulic parameters for the van Genuchten equation were determined using the inverse optimisation procedure in Hydrus 1D (*Šimůnek et al., 2008*). Water profile data from the soil columns in Set A were used, after removing obvious outliers in the data that were determined to be due to soil loss during column sampling (*Hopmans et al., 2002*). The initial water content was set to 0.15 ($cm^3\ cm^{-3}$), the measured water content of the repacked columns, for all optimisations. Free water absorption was simulated by applying constant water content boundary condition of 0.64 ($cm^3\ cm^{-3}$) to the opening of the column. The lower boundary condition was set to free drainage. The maximum number of iterations was set to 50 and 86 water content points across five time steps were used in the inverse scenario from the Set A column experiments. Weighting of inverse data was by standard deviation and an equal weighting was applied to all the water content values. Residual soil water content ($\theta_r$), $\theta_s$, and $K_{sat}$ were set or optimised depending on the optimisation scenario (*Fit All* is the term applied when all parameters were optimised and *Set Measured* is used when $\theta_r$, $\theta_s$, and $K_{sat}$ were set to independently measured values).

The remaining empirical parameters, α, $n$, and $l$ were fitted by running the inverse parameter estimation option in Hydrus 1D. The initial values of α, $n$, and $l$ were based on the default values given for the Loam soil in Hydrus 1D (α = 1.56, $n$ = 0.173, and $l$ = 0.5).

Initial estimates for $θ_r$ and $θ_s$ were 0.05 and 0.58 (cm$^3$ cm$^{-3}$) and $K_{sat}$ was set to 0.10 cm min$^{-1}$. Saturated soil water content ($θ_s$) and $K_{sat}$ values were independently measured in falling head $K_{sat}$ experiments (*Reynolds et al., 2002*). A column of water (4.2 cm in diameter and 12 cm high) was applied to wet repacked soil core packed to a bulk density of 1.0 with air dry soil sieved to <2 mm. The soil cores were two cm high and had an internal diameter the same as the water column sitting above it. Triplicate measurements of conductivity were recorded on four separate cores. Residual soil water content ($θ_r$) was estimated based on the air-dry soil water content.

The Rosetta PTF model (*Schaap, Leij & Van Genuchten, 2001*) was used to estimate soil hydraulic properties as a comparison against the inverse modelling method. The soil particle size measurements (Table 1) and bulk density (1.03 g cm$^{-3}$) were used to estimate soil hydraulic parameters. In a second prediction, moisture retention at −33 and −1,500 kPa (0.34 and 0.22 cm$^3$ cm$^{-3}$, respectively) were also included as inputs into Rosetta. The water content at −33 kPa was determined on a suction table apparatus and −1,500 kPa were determined using pressure plate apparatus (*Cresswell, 2002*).

## Modelling water and solute absorption in soil wedges

Hydrus 2D/3D was used to model water distribution in the horizontal column experiments based on the same initial and boundary conditions used in Hydrus 1D during inverse optimisation. Horizontal flow in Hydrus 2D/3D was simulated by setting the geometry to a 2D horizontal plane. The geometry of the flow domain was set to a column two cm in diameter and 50 cm long. Soil hydraulic parameters having the lowest values for the objective function were used to simulate water absorption. Nitrate absorption was predicted by applying a third-type (Cauchy) solute boundary at the inlet of the column ($x$ = 0 cm) at a constant concentration of 110 μmol NO$_3^-$ cm$^{-3}$. Bulk density was set to the measured value of 1.03 g cm$^{-3}$, longitudinal and transverse dispersivities were set to 0.3 and 0.03 cm, respectively (*Ajdary et al., 2007*), and the diffusion coefficient was neglected as it was considered negligible relative to the dispersion (*Hanson, Šimůnek & Hopmans, 2006*; *Ajdary et al., 2007*). Parameters for the Langmuir equation (Eq. (1)) to describe NO$_3^-$ adsorption were determined from the data measured in column Set B.

## Modelling water and solute in horizontal columns

To simulate water and NO$_3^-$ distribution in the wedge experiments, a 40 × 40 cm flow domain was created in a 2D axisymmetrical vertical flow geometry. The infiltration point at five cm depth was represented by a semicircle with three cm radius. The flux from the source was 10.61 cm h$^{-1}$ (Eq. (4)), equivalent to a dripper output of 1,200 cm$^3$ h$^{-1}$.

$$σ = \frac{Q}{4\,π\,r^2}\,,$$

(4)

**Table 2** Parameter estimation results for the *Fit All* and *Set Measured* parameter sets.

| Inverse scenario | $\theta_r$ (cm$^3$ cm$^{-3}$) | $\theta_s$ (cm$^3$ cm$^{-3}$) | $\alpha$ (cm$^{-1}$) | $n$ | $K_{sat}$ (LT$^{-1}$) | $l$ | $\Phi$ |
|---|---|---|---|---|---|---|---|
| *Fit All* | 0.112 (0.247) | 0.560 (0.007) | 0.036 (0.006) | 2.030 (0.684) | 0.115 (0.061) | 3.847 (5.442) | 0.014 |
| *Set Measured* | 0.054 (0.003*) | 0.580 (0.04*) | 0.038 (0.005) | 2.335 (0.279) | 0.104 (0.029*) | 3.175 (0.578) | 0.020 |

Notes:

Values in parentheses show the 95% confidence intervals of the estimated parameters. $\Phi$ indicates the value of the objective function.

* Independently measured.

where $\sigma$ is the flux from the surface of the source (cm h$^{-1}$), $Q$ is the total volumetric flux (cm$^3$ h$^{-1}$), and $r$ is the radius of the spherical source (cm).

The finite element mesh of the flow domain, which determines the level of model resolution in the calculations, was set to 0.5 cm in both $z$ and $h$ directions. No flux was allowed through the column boundaries. The infiltration source was set as a variable flux boundary so that water and solute applications could be controlled according to the two irrigation scenarios described for the wedge columns (Fig. 2). Nitrate absorption was predicted in the same way as for the horizontal columns. The time-variable boundary condition was used to apply the solute (110 $\mu$mol$_c$ NO$_3^-$ cm$^{-3}$) only for the first 0.5 h of water application to the column.

## Statistical analysis

The root mean square error (RMSE) was calculated as the error between the measured and simulated water content and NO$_3^-$ concentrations. Comparisons of the RMSE values with predictions from different parameter sets allowed those that produced the lowest errors to be identified. Comparisons of simulated RMSE values with those calculated from measured data allowed the significance of the model error to be assessed in relation to measurement error. This method has been commonly used to measure the quality of model predictions in previous studies (*Skaggs et al., 2004*; *Ajdary et al., 2007*; *Arbat et al., 2008*; *Patel & Rajput, 2008*). Part of the inverse modelling in Hydrus 1D allows calculation of a correlation matrix that specifies the correlation between the fitted coefficients and statistical information about the fitted parameters.

## RESULTS

### Parameter determination and model validation in horizontal columns

Values for soil hydraulic parameters estimated from column Set A are presented in Table 2. The two inverse scenarios produced slightly differing values, with the *Fit All* parameters having a lower value for the objective function $\Phi$ than the *Set Measured* suggesting that the former gave a slightly closer fit between the predicted and measured soil water profiles. The 95% CIs for the optimised values for $\alpha$, $n$, and $l$ where smaller compared to the values when all parameters were optimised. The parameters in the *Fit All* scenario had higher uncertainty and greater correlation between fitted parameters compared to the *Set Measured* parameters (Tables 2 and 3). The correlation matrix shows there were three high values in the *Fit All* parameter function compared to one in the *Set Measured* results; $\alpha$ and $n$ being highly correlated (Table 3, bold entries). High correlation values

**Table 3 Correlation matrix of the inverse function.**

| Fit All | $\theta_r$ | $\theta_s$ | $\alpha$ | $n$ | $K_{sat}$ | $l$ |
|---|---|---|---|---|---|---|
| $\theta_r$ | 1.000 | | | | | |
| $\theta_s$ | 0.016 | 1.000 | | | | |
| $\alpha$ | **0.977** | −0.046 | 1.000 | | | |
| $n$ | 0.656 | −0.243 | 0.561 | 1.000 | | |
| $K_{sat}$ | −0.753 | 0.118 | −0.649 | **−0.976** | 1.000 | |
| $l$ | **−0.975** | −0.149 | **−0.941** | −0.605 | 0.737 | 1.000 |

| Set Measured | $\alpha$ | $n$ | $l$ |
|---|---|---|---|
| $\alpha$ | 1.000 | | |
| $n$ | **0.937** | 1.000 | |
| $l$ | −0.283 | 0.063 | 1.000 |

(magnitude >0.9) indicate parameter nonuniqueness and a correspondingly high uncertainty (*Hopmans et al., 2002*; *Šimůnek & Van Genuchten, 1996*). The optimised value of $\theta_r$ was 0.112 and the 95% CI that ranged from −0.135 to 0.359, which includes the measured value (0.054 ± 0.003). Saturated water content ($\theta_s$) was 0.56 (0.553–0.567) and was significantly different from the independently measured values (0.58 ± 0.04; Table 2). The measured $K_{sat}$ (0.104 ± 0.0299) fall within the 95% CI [0.054–0.176; mean best fit value of 0.115] of the optimised value.

The two parameter sets were tested against independently measured data from the horizontal columns (Column data Set B) and the point-source wedge experiments. Predicted water distributions and measured water content in the horizontal columns from column Set B are shown in Fig. 3A. We have plotted θ and $NO_3^-$ profiles against the Boltzmann variable $X$ (distance/√time; cm s$^{-1/2}$; *Smiles et al., 1978*; *Phillips & Bond, 1989*). Both parameter sets showed similarly good correspondence to the measured data (continuous and dotted lines), which is confirmed by the RMSE and $R^2$ values (Table 4). Accurate predictions of θ profiles after absorbing water for 80 and 320 min are not surprising given that data from Set A (27–70 min) are not statistically different from Set B when normalised against the Boltzmann variable. The results, however, do show that under the unsaturated absorption scenario, predictions made for longer times (80 and 320 min) are still accurate when the predictions are extended beyond the range of optimised data. The consistency between predictions based on short time θ and $NO_3^-$ data (Set A) and longer time θ and $NO_3^-$ data (Set B) further demonstrate the experiments had a good degree of repeatability and produced consistent data across a range of time scales.

In comparison, predicted water content using soil hydraulic parameters determined by Rosetta are shown in Fig. 4. These data show the piston front to be significantly behind the measured data (combined data from column Set A and B) especially when the water content at −33 and −1,500 kPa are included in the model. Removal of the FeO before determining the sand, silt, and clay content did not improve the predictions of water retention parameters or saturated hydraulic conductivity. Deriving the water retention

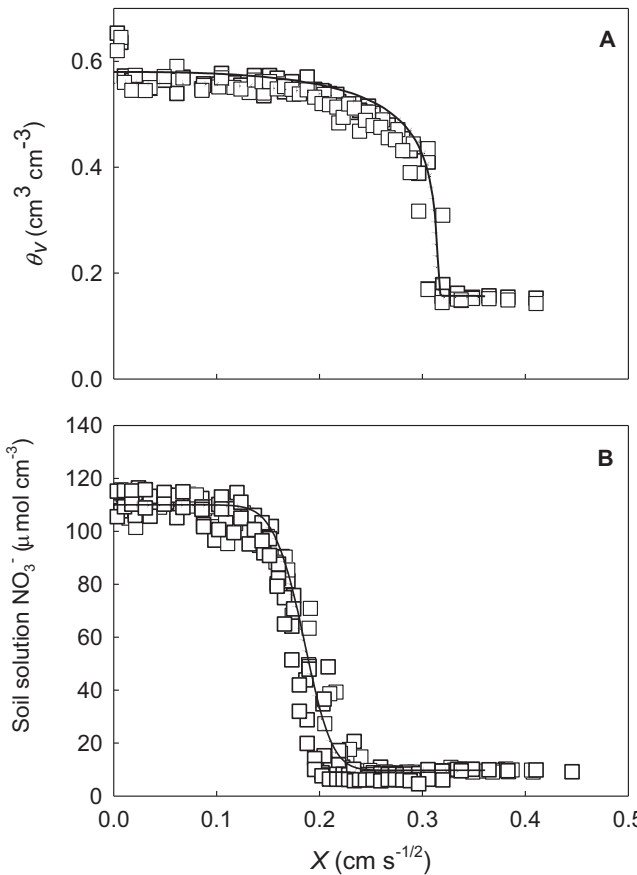

**Figure 3** (A) Fits of the *Fit All* (solid line) and *Set Measured* (dotted line) parameters to the measured water profile data (squares) from column Set B not included in the inverse optimisation. (B) Fits for soil solution $NO_3^-$.

**Table 4** Root mean square error (RMSE) of water and $NO_3^-$ profiles determined using the *Fit All* and *Set Measured* parameters in comparison to the measured data in the horizontal column experiments presented in Fig. 3.

| Inverse scenario | $\theta_v$ (cm$^3$ cm$^{-3}$) | | $NO_3^-$ ($\mu mol_c$ cm$^{-3}$ soln) | |
|---|---|---|---|---|
| | RMSE | $R^2$ | RMSE | $R^2$ |
| *Fit All* | 0.04 | 0.89 | 7.50 | 0.97 |
| *Set Measured* | 0.05 | 0.91 | 7.96 | 0.97 |
| Measured[†] | 0.04 | | 6.19 | |

**Note:**
[†] "Measured" indicates the variation in the measured data calculated from the second set of horizontal soil columns where two $NO_3^-$ and water measurements were made at identical points and times.

parameters and saturated hydraulic conductivity with Rosetta, values that use PTFs to predict the Van Genuchten parameters, produced significant inaccuracies in the predictions (see Fig. 4). In these experiments they are certainly less accurate than parameters derive from inverse modelling (Fig. 3).

The predicted $NO_3^-$ distribution is shown in Fig. 3B. The measured and predicted $NO_3^-$ distributions were compared from both Set A (27–70 min) and Set B

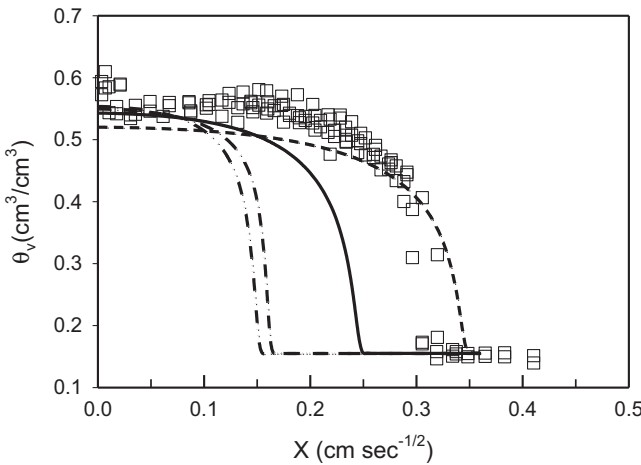

**Figure 4 Predicted volumetric water content using Rosetta derived soil hydraulic parameters and measured water profile data (squares) against the Boltzmann variable ($X$).** The *solid* line represents derived parameters using particle size data measured after removal of iron oxides and no water content data; the *dash* line represents parameters predicted using particle size without removing iron oxides and without water content data; the *dash-dot–dash* line refers to parameter predicted using particle size without removing iron oxides and with soil water content at −33 and −1,500 kPa; the *dash-dot-dot–dash* line refers to parameters predicted using particle size after removing iron oxides and with soil water content at −33 and −1,500 kPa.

(80 and 320 min) because the $NO_3^-$ distribution was not used in the inverse optimisation. The good prediction of $NO_3^-$ distribution by the two parameter sets, when the Langmuir isotherm parameters were included in the simulations, is confirmed by the $R^2$ and RMSE values (Table 4). Due to the inaccuracy of water content prediction using Rosetta, no $NO_3^-$ data are presented.

## Model validation using 3D wedge infiltration

The predicted distributions of water and $NO_3^-$ throughout the soil wedge after the two irrigation scenarios are shown in Fig. 5. This figure also shows the positions of horizontal and vertical transects presented in Figs. 6 and 7. These figures show the agreement between the measured and predicted water and $NO_3^-$ profiles in the wedge. Comparisons of the RMSE and $R^2$ calculations indicated that both the *Fit All* and *Set Measured* parameter sets predicted very similar distributions, although the *Fit All* parameters produced slightly better predictions of $NO_3^-$ distribution in the longer irrigation scenario. The measured RMSE, calculated from columns where duplicate measurements were taken at identical times and locations, were similar to the RMSE of the predicted values (Table 5). This indicates that the errors between the measured and predicted values were very similar to the errors of replicate measurements at identical points and times in the wedge experiments. The predictions achieved using the two parameter sets were therefore considered to be suitable to estimate water and $NO_3^-$ distribution in the point-source flow scenario of the wedge column. The "*Set Measured*" parameters are preferred because there is less auto correlation between the fitted parameters ($\alpha$, *n*, *l*).

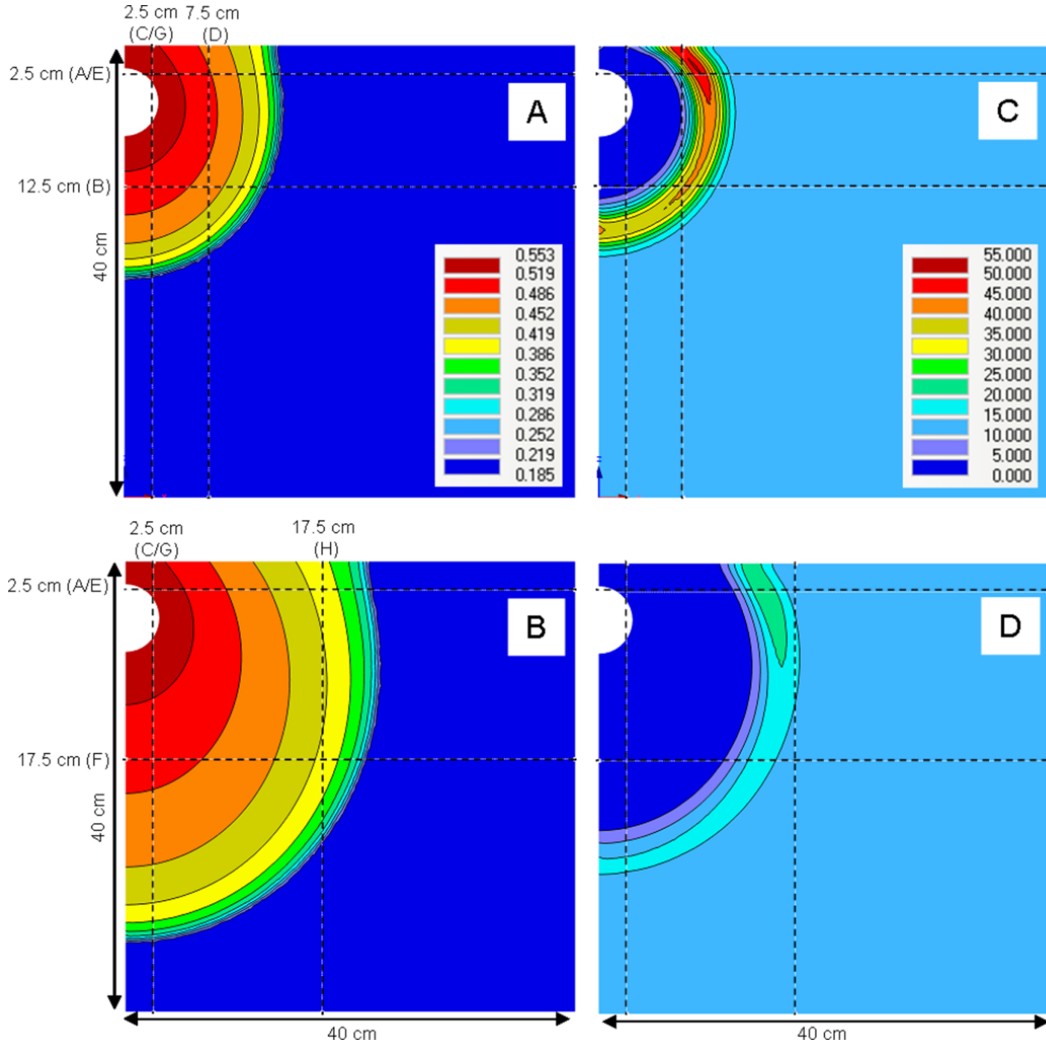

**Figure 5 Estimation of water and NO$_3^-$ in the wedge experiment scenario using the *Set Measured* parameters.** (A) and (C) show water content (cm$^3$ cm$^{-3}$) and NO$_3^-$ (μmol cm$^{-3}$), respectively, for irrigation scenario A (2 h experiment, Fig. 2). (B) and (D) show water content (cm$^3$ cm$^{-3}$) and NO$_3^-$ (μmol cm$^{-3}$), respectively, for irrigation scenario B (24 h experiment, Fig. 2). Horizontal and vertical transects and their symbols correspond to the water and solute profile plots in Figs. 6 and 7.

## DISCUSSION

Inverse optimisation using Hydrus-1D can be used to effectively estimate soil hydraulic properties from simple 1D flow experiments. The flow parameters derived from 1D columns were suitable for describing flow under more complex 3D flow scenarios. These results show that the use of absorption columns offers an alternative to PTF methods and has the advantage that the parameters are determined using data from the actual soil type under consideration.

Comparisons of optimised parameters with independently measured data in the horizontal column showed that the fitted parameters were able to accurately predict water distribution in these flow scenarios. Further, water distribution could be accurately

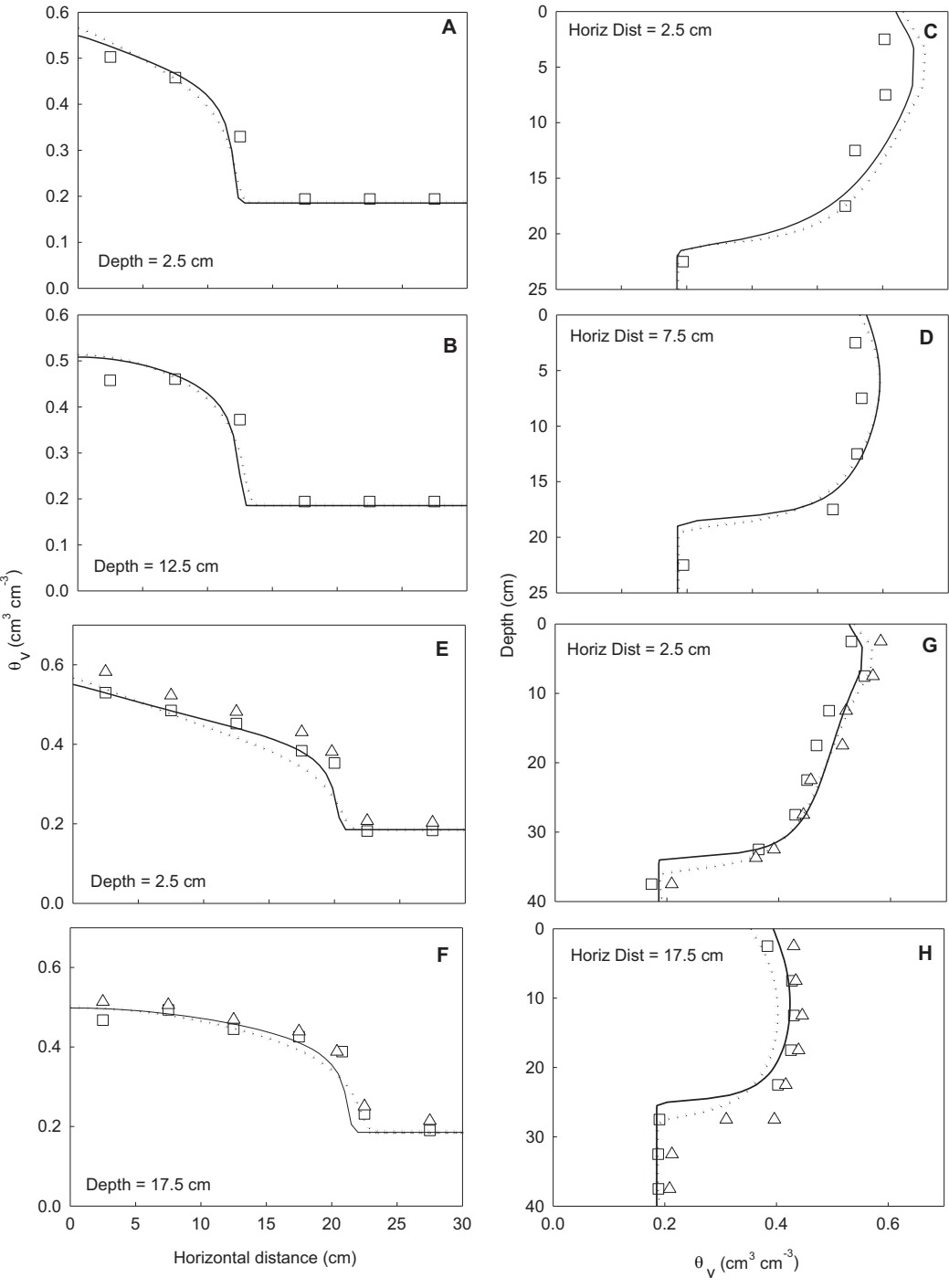

**Figure 6 Horizontal (A, B, E, F) and vertical transects (C, D, G, H) of water content (cm³ cm⁻³) in the wedge columns.** The symbols indicate measured data (squares represent replicate one and triangles replicate two). Solid lines represent simulations using the Fit All parameters and the dotted lines simulations using the Set Measured parameters. A to D show transects from irrigation scenario A (2 h experiment, Fig. 2) and E to H show transects from irrigation scenario B (24 h experiment, Fig. 2). Location of transects is shown in Fig. 5. The profiles were taken at same times as snapshots shown in Fig. 5.

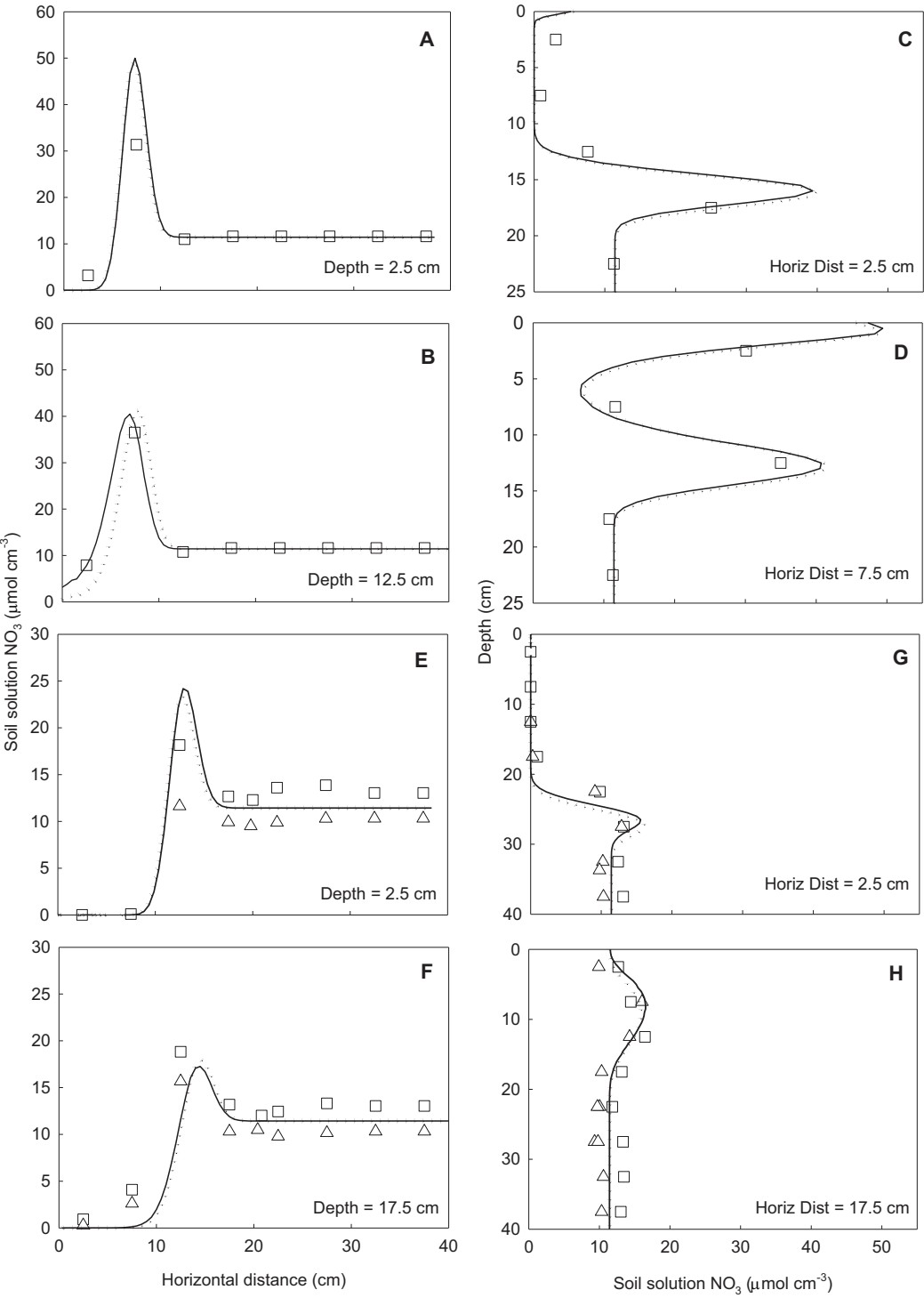

**Figure 7 Horizontal (A, B, E, F) and vertical (C, D, G, H) transects of soil solution $NO_3^-$ concentration ($\mu mol_c$ $cm^{-3}$) in the wedge columns.** Symbols indicate measured data (squares represent replicate one and triangles replicate two). Solid lines represent simulations using the *Fit All* parameters and dotted lines simulations using the *Set Measured* parameters. The $NO_3^-$ reaction parameters were included in all simulations. Key to panels and transects same as for Fig. 5. The profiles were taken at same times as snapshots shown in Fig. 5.

**Table 5 Root mean square error (RMSE) of the fit of the two parameter sets used to predict water and NO$_3^-$ distribution in the wedge experiments.**

| Parameters | Irrigation treatment | $\theta_v$ (cm$^3$ cm$^{-3}$) | | NO$_3^-$ ($\mu$mol$_c$ cm$^{-3}$ solution) | |
|---|---|---|---|---|---|
| | | RMSE | $R^2$ | RMSE | $R^2$ |
| *Fit All* | Scenario A | 0.04 | 0.94 | 2.40 | 0.95 |
| | Scenario B | 0.03 | 0.97 | 3.05 | 0.81 |
| *Set Measured* | Scenario A | 0.04 | 0.96 | 2.16 | 0.95 |
| | Scenario B | 0.02 | 0.98 | 3.43 | 0.76 |
| Measured[†] | | 0.03 | | 2.64 | |

**Note:**
[†] "Measured" RMSE values indicate the variation in the measured data calculated from the wedge experiments from Irrigation Scenario B columns where two NO$_3^-$ and water measurements were made at identical points in the wedges.

predicted for absorption periods four times longer than the data used in the inverse optimisation. This provides preliminary evidence that the parameters can predict water distribution outside the range of fitted values. However, this result is not surprising because the measured water content profiles coalesce to a single curve when presented against the Boltzmann variable $X$ (cm s$^{-1/2}$). Further evidence for this was provided by testing the parameters in the alternative point-source 3D flow scenario. The results presented in this paper demonstrate there is scope to use soil hydraulic parameters obtained from simple horizontal absorption experiments to accurately estimate water flow under more complex 3D conditions in uniform repack soil conditions (isotropic).

The inverse optimisations in this study produced two parameter sets that were capable of providing good predictions of water flow in the two flow scenarios. *Hopmans et al., (2002)* suggests that if various parameter sets produce similar model outcomes, the soil hydraulic parameters may be unidentifiable and the inverse optimisation may be ill-posed. However, our data (Table 2) shows that the values identified in the two scenarios are within the 95% CI estimates of the predictions; the difference between the parameter sets are therefore not significant. Limiting the number of parameters optimised in the inverse procedure reduced the uncertainty of the fitted parameters without significantly affecting the accuracy of model predictions. Further, the *Fit All* scenario gave high correlations of three parameters in comparison to the one high value for the *Set Measured* scenario (Table 3). Limiting the number of parameters in the inverse scenario was shown to be advantageous because parameter variation was reduced. These findings are consistent with the recommendations of *Hopmans et al. (2002)* but contrast with the study of *Sonnleitner, Abbaspour & Schulin (2003)*. The latter work indicated that maximising the number of variables in the inverse optimisation increased the ability of parameters to describe water flow in alternative scenarios.

In our wedge study, only minor differences were observed between predictions when the number of optimised parameters was reduced. Furthermore, reducing the number of parameters in the inverse optimisation was advantageous because parameter uncertainty was reduced (Table 2). Our results show that, where practical, there is benefit in conducting additional measurements to estimate $\theta_s$ and $K_{sat}$, which are two of the most sensitive parameters of the model (*Arbat et al., 2008*). The benefits of using measured
parameters, in combination with inverse modelling of water content data, has also been demonstrated by *Kandelous et al. (2011)*. In these columns, hydraulic conductivity is not independently measured; rather sorptivity is measured and the hydraulic conductivity must be inferred with a model.

Including solute reaction parameters enables the Hydrus model to accurately predict reactive solute distribution in the soil. The retardation in the $NO_3^-$ relative to the inflowing water indicates that the solute was adsorbed by the soil. Sorption of $NO_3^-$ was included in Hydrus by using the Langmuir equation to approximate the partitioning of $NO_3^-$ between the soil solution and the adsorbed phases. The Langmuir equation has been used previously to describe solute adsorption in soil (*Katou, Clothier & Green, 1996*; *Qafoku, Sumner & Radcliffe, 2000*; *Phillips, 2006*).

The distribution of solutes adsorbed to soil during water flow has been simulated under point-source infiltration in previous studies using Hydrus 2D/3D (*Hanson, Šimůnek & Hopmans, 2006*). However, validation under these flow scenarios has received little attention. *Ben-Gal & Dudley (2003)* observed that predictions of reactive P transport from a drip irrigation system showed a similar distribution to measured data, but they did not make any statistical comparisons. Reactive solute transport was previously validated under other flow scenarios (*Persicani, 1995*; *Moradi, Abbaspour & Afyuni, 2005*). Our results validate the inclusion of the Langmuir equation in Hydrus for the prediction of reactive solute movement for 1D and 3D flow conditions. Furthermore, the results show that reactive solute parameters determined from relatively simple 1D adsorption columns can be used to accurately predict solute distribution under 3D conditions.

Other studies that have used Hydrus to predict water movement through soils have utilised PTFs to estimate soil hydraulic parameters (*Epino et al., 1996*; *Skaggs et al., 2004*; *Li, Zhang & Rao, 2005*; *Phillips, 2006*; *Siyal & Skaggs, 2009*). The suitability of parameters predicted by PTFs relies on the amount of data collected from soils with similar particle size distribution, bulk density, and water-holding capacity. We investigated use of the Rosetta model (*Schaap, Leij & Van Genuchten, 2001*) to obtain parameters for the same Red Ferrosol used here, but it provided less accurate estimates of water distribution in comparison to parameters determined from inverse modelling (Fig. 4). The high value of the pore connectivity parameter, l (Table 2), that resulted from inverse optimisation is in contrast to the value of 0.5 used on the Rosetta model (*Cook & Cresswell, 2007*). This difference may explain the limitations of Rosetta to accurately predict water flow in the repacked columns in these particular experiments.

This finding contrasts with those of *Kandelous & Šimůnek (2010a)* where parameters estimated from Rosetta produced acceptable predictions of water movement in laboratory studies. The differing results in our study may be in part due to the limited data for Australian Red Ferrosols available in the Rosetta soil database. In general, this paper confirms that inverse optimisation is advantageous provided enough data has been collected over a sufficient range of water contents (*Šimůnek et al., 2000*; *Sonnleitner, Abbaspour & Schulin, 2003*). The use of inverse optimisation applied to horizontal infiltration columns provides a simple technique to accurately determine reaction parameters.

If parameters determined from inverse optimisation are to successfully describe water flow in alternative scenarios the soil properties must be the same. This was achieved in our laboratory because careful packing was possible in both the horizontal columns and soil wedges. For a field scenario, a similar method of predicting water and solute flow would need laboratory experiments on undisturbed cores. Similarly, *Kandelous & Šimůnek (2010a)* found that parameters suitable for describing water movement in packed laboratory columns were not capable of describing water distribution in an undisturbed field soil.

## CONCLUSION

Inverse modelling procedures in Hydrus confirm that soil hydraulic parameters can be reliably obtained from simple 1D diffusive water uptake soil column studies. The derived parameters are capable of accurately describing diffusive water movement over extended times and in alternative dynamic flow scenarios to those in which they were fitted. These results demonstrate that simple water uptake column experiments can be used to provide suitable flow conditions for accurate determination of reaction parameters under dynamic flow conditions. Reducing the number of parameters in the optimisation procedures by imposing independently measured values for $\theta_s$, $\theta_r$, and $K_{sat}$ decreased parameter uncertainty (or increased parameter uniqueness) without significantly impacting the accuracy of model predictions. These results show there is merit in pursuing this method in more complex scenarios since it may provide a simpler and cheaper way of determining hydraulic parameters in field conditions.

Solution $NO_3^-$ and adsorbed $NO_3^-$ concentrations collected from a combined water uptake-$NO_3^-$ tracer test provided the data to fit the reaction parameters for the Langmuir isotherm, which in turn were included in the Hydrus model to predict reactive solute transport. We have demonstrated the ability of HYDRUS to integrate unsaturated flow processes and independently determined reactive transport processes based on independent experiments involving the complex interplay of dynamic flow and reactive transport.

## ACKNOWLEDGEMENTS

We thank Dr Freeman Cook for his assistance with Hydrus modelling and Dr David Smiles for providing his expertise in experimental design and analysis.

### Funding

An Australian Federal Government Postgraduate Scholarship supported Dr. Kirkam's research. The funders had no role in study design, data collection and analysis, decision to publish, or preparation of the manuscript.

### Grant Disclosure

The following grant information was disclosed by the authors:
Australian Federal Government Postgraduate Scholarship.

## Competing Interests

The authors declare that they have no competing interests.

## Author Contributions

- James M. Kirkham conceived and designed the experiments, performed the experiments, analyzed the data, contributed reagents/materials/analysis tools, prepared figures and/or tables, authored or reviewed drafts of the paper, approved the final draft.
- Christopher J. Smith conceived and designed the experiments, analyzed the data, contributed reagents/materials/analysis tools, authored or reviewed drafts of the paper, approved the final draft.
- Richard B. Doyle conceived and designed the experiments, authored or reviewed drafts of the paper, approved the final draft.
- Philip H. Brown conceived and designed the experiments, authored or reviewed drafts of the paper, approved the final draft.

## Data Availability

Raw data is available as Supplemental Files.

## Supplemental Information

Supplemental information for this article can be found online at http://dx.doi.org/10.7717/peerj.6002#supplemental-information.

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
