# Peer review of "Inverse modelling for predicting both water and nitrate movement in a structured-clay soil (Red Ferrosol)"

_PeerJ, doi:10.7717/peerj.6002_

## Round 0.1 · original submission · Minor Revisions

The manuscript was evaluated by two reviewers, who were quite positive about your work. Both ask for some more details and supplementary information, in particular related to the modeling.

I am looking forward to see the revised version of the manuscript.

Reviewer 1 ·

Basic reporting

This paper is well written and provides interesting results on the use of inverse modelling to parameterise models for water flow and solute transport.

Detailed suggestions for consideration by the authors have been added as comment boxes throughout the pdf manuscript.

Experimental design

A critical component of this paper is the inverse modelling of soil hydraulic parameter. However, the explanation and detail of how the column data was incorporated and run in Hydrus 1-D to generate the inverse parameters is insufficient for readers to easily understand how this was undertaken.

In addition a table of the column data that was used in the inverse modelling would be useful to enable others to repeat the parameter optimisation. If this is felt to be too big to go into the paper then a much clearer table needs to be added in the supplemental files.

The paper reports on a small study, with little replication, however the excellent fit between measured and modelled data suggests that, unless the authors have been very lucky, further replication would probably have been superfluous. The controlled conditions where soil structure was accurately reproduced also added to the success of this small study.

The authors have provide a great deal of information in the supplemental files, however, more descriptive metadata would be useful if they are to be readily accessible to readers.

Validity of the findings

The findings are generally well described and supported by the data. Some extra discussion on the authors thoughts on the practical merits of having to include independent measurement for the inverse modelling, and why adding retention points might make the Rosetta results worse would be interesting.

Additional comments

I look forward to the review comments being addressed and seeing the paper published.

Annotated reviews are not available for download in order to protect the identity of reviewers who chose to remain anonymous.

·

Basic reporting

This is a generally well written manuscript. There are some aspect where I think the authors need to explain or describe what they have done in more detail.

Experimental design

The value of l in Table 2 is quite different to the standard HYDRUS value of 0.5, yet no attention or discussion about this occurs. Having a large value of l will mean that the hydraulic conductivity decreases rapidly with water content. All the Rosetta derived values will have used 0.5, rather than values > 3 as given in Table 2. This will have a large effect on the shape of figure 3a, which is why the Rosetta derived values may have been unable to match the experimental data in figure 4. The authors show see what happens if the alter l with the other Rosetta values. Cook and Cresswell (2008) (Cook FJ and Cresswell HP (2007). Chapter 84 Estimation of Soil Hydraulic Properties. In Soil Sampling and Methods of Analysis. M.R. Carter and E.G. Gregorich (Eds.), Canadian Society of Soil Science, Taylor and Francis, LLC, Boca Raton, Fl, 1139-1161.) discuss how to estimate l.

Validity of the findings

Thevan Genuchten equation and Mualem equations need to be shown. The values of the parameters in this equation are shown in Table 2 but no explanation of what the parameters represent. This could be done either in the text or in the caption of Table 2. The symbols in Table 2 need correcting as they differ from Table 3 and the rest of the text and figures

Additional comments

The referencing needs to be fixed with citations missing from the reference list and references which are not cited. There are also a few minor grammatical corrections required.

Minor Corrections.
1. Lines 48 and 49 and throughout the rest of the manuscript. Vrugy and Bouten (2002) and Wohling et al. (2008) are not in the reference list. Also for the Wohling reference et al. is not italicised
2. Line 62. I think the reference to Mallant et al. (2007) should be Mallants et al. (2007).
3. Line 77. Jacques et al. (2012) is not in the reference list.
4. Line 171. The abbreviation CI is not defined.
5. Line 201. Replace ‘was’ with ‘were’.
6. Line 208. I do not know what ‘equivalent’ refers to?
7. Line 219. Replace ‘Horizontal Columns’ with ‘Wedges’.
8. Line 255. Simunek and van Genuchten (1996) is not in the reference list.
9. Line 323. I think Hopmans (2002) should be Hopmans et al. (2002).
10. Line 428. Gardenas et al. is not cited.
11. Lines 428-487. Romos et al. (2011, 2012) are not cited.
12. Line 520. Soil Survey Staff (1992) is not cited.
13. Table 2. Symbols used are not correct.

---

## Round 0.2 · accepted · Accept

Thank you for your hard work on the revision, I am happy to accept your manuscript for publication.